# C-Terminal Extended Domain-Independent Telomere Maintenance: Modeling the Function of TIN2 Isoforms in *Mus musculus*

**DOI:** 10.3390/ijms26062414

**Published:** 2025-03-07

**Authors:** Chiao-Ming Huang, Yi-Ling Shen, Chia-Lo Ho, Tzeng-Erh Chen, Hsuan-Yu Hsia, Zhou Songyang, Liuh-Yow Chen

**Affiliations:** 1Molecular and Cell Biology, Taiwan International Graduate Program, Academia Sinica and Graduate Institute of Life Sciences, National Defense Medical Center, Taipei 11490, Taiwan; iointheair@gate.sinica.edu.tw; 2Institute of Molecular Biology, Academia Sinica, Taipei 11529, Taiwan; ling0725@gate.sinica.edu.tw (Y.-L.S.); eggcookie@gap.kmu.edu.tw (C.-L.H.); monkichi0802@gmail.com (T.-E.C.); hcu1228@gate.sinica.edu.tw (H.-Y.H.); 3Sun Yat-Sen Memorial Hospital, Sun Yat-Sen University, Guangzhou 510275, China; songyang@bcm.edu

**Keywords:** telomere, alternative splicing, DNA damage

## Abstract

TIN2 (TERF1 interacting nuclear factor 2) is a telomeric shelterin complex component, essential for telomere protection and early embryonic development in mammals. In humans, TIN2 isoforms arise from alternative splicing, but their specific roles in vivo remain unclear. Here, we explore TIN2 isoform functions in the laboratory mouse *Mus musculus*. Our comparative analysis of TIN2 protein sequences reveals that mouse TIN2 (TINF2) closely resembles the human TIN2L isoform, both of which harbor a C-terminal extended domain (CTED) absent from the human TIN2 small (TIN2S) isoform. To further characterize the functions of TIN2 isoforms, we generated a *Tinf2* LD (long-form deficiency) allele in *M. musculus* encoding a short form of TINF2 lacking the CTED. Mice heterozygous or homozygous for this *Tinf2* LD allele were viable, fertile, and showed no tissue abnormalities. Furthermore, protein product of *Tinf2* LD allele localized to telomeres and maintained telomere integrity in mouse embryonic fibroblasts, demonstrating that the CTED is dispensable for telomere protection and normal development in mice. These findings indicate functional redundancy among TIN2 isoforms and underscore the utility of the *Tinf2* LD model for uncovering isoform-specific mechanisms of telomere regulation.

## 1. Introduction

Positioned at the ends of linear chromosomes, telomeres consist of repetitive DNA sequences that are protected by protein components. Telomere malfunction, often caused by mutations in telomeric proteins, leads to disease phenotypes resembling premature aging [1]. In terms of cellular phenotypes, telomeric proteins protect chromosome ends from being mistakenly identified as double-strand breaks (DSBs) and consequent phenotypic manifestations, including the fusion of chromosome ends. The well-characterized telomere-protecting protein complex, i.e., the telomere shelterin complex, comprises six essential proteins: telomeric repeat binding factor 1 (TERF1, also referred to as TRF1); telomeric repeat binding factor 2 (TERF2, also called TRF2); TERF1 interacting nuclear factor 2 (TINF2, commonly known as TIN2); TERF2 interacting protein (TERF2IP, known as RAP1); adrenocortical dysplasia (ACD, with TPP1 being another commonly used name for ACD); and the protection of telomeres 1 (POT1) [2,3,4]. The depletion of any of these telomere shelterin components was found to trigger a telomere damage response [5,6,7] and lead to telomere fusion [8,9,10].

TIN2, a key component of the shelterin complex, plays a pivotal role in protecting mammalian cells from telomeric damage [11,12,13,14,15]. Its role in maintaining telomeres hinges on its function as a scaffold protein for direct interactions with TRF1, TRF2, and TPP1 [13]. Disruption of the interactions between TIN2 and these other shelterin components can trigger a telomeric DNA damage response and fusion events [13]. Additionally, TIN2 is crucial for telomerase recruitment [16,17], and it has been implicated in the establishment of heterochromatin at telomeric regions [18,19]. Mutations in *TINF2*, the gene encoding TIN2, have been linked to telomere-related disorders such as dyskeratosis congenita [20,21,22], Revesz syndrome [23], and Hoyeraal–Hreidarsson syndrome [24]. These mutations are often associated with dysregulated telomere length, highlighting TIN2’s critical role in telomere length homeostasis [25,26,27]. In mice, deletion of the TIN2 gene (*Tinf2*) has been shown to impede blastocyst formation, resulting in early embryonic lethality [28]. Mutant mice hosting a *Tinf2* allele reflecting a dyskeratosis congenital disease mutation (i.e., the *Tinf2* DC allele) exhibit homozygous lethality, with heteroxygous TIN2^DC/+^ individuals displaying decreased fertility, shortened telomeres, and pancytopenia [29]. These findings highlight essential roles for TIN2 in telomere maintenance and overall organism physiology.

Three TIN2 isoforms—TIN2S (small), TIN2M (medium), and TIN2L (long)—arise from alternatively spliced TIN2 transcripts of the *TINF2* gene in humans [30,31,32]. These isoforms share identical sequences for the N-terminal 1–354 residues, but TIN2L possesses a C-terminal extended domain (CTED) not present in TIN2S or TIN2M. All three isoforms localize to telomeres and contain domains that interact with TRF1, TRF2, and TPP1. Additionally, TIN2L may be associated with the nuclear matrix, implying a distinct role for it among the three isoforms [30,33]. Interestingly, the C-terminus of TIN2L contains a phosphorylation site at the serine-396 residue that, when phosphorylated by casein kinase 2 (CK2), may enhance its interaction with TRF2. This phosphorylation event could facilitate the formation of a more cohesive shelterin complex structure, potentially strengthening telomere protection [32]. Thus, the distinct TIN2 isoforms may give rise to different shelterin structures, potentially contributing to diverse mechanisms of telomere maintenance. Notably, an allele hosting a frameshift TIN2 mutation, which encodes a mutant TIN2 protein lacking the CTED and TRF1 binding site, induced dysregulated telomere length in RPE1-hTERT cells [26], suggesting that the CTED may exert unique roles in telomere maintenance.

In contrast to humans, the alternative splicing of TIN2 has not been observed in the widely used laboratory mouse *Mus musculus*, though it has being reported for another mouse species, such as *Mus spretus* [31]. Comparative protein sequence analysis indicates that mouse TINF2 is similar to human TIN2L, with the C-terminal domain of mouse TINF2 being analogous to the extended region of human TIN2L. Based on this information, we developed a genetically modified *Mus musculus* model to express a truncated TINF2 that mimics the human TIN2S variant lacking the CTED. This model allowed us to investigate TIN2 isoform functionality in vivo. However, our results indicate that CTED deletion from TINF2 did not result in developmental abnormalities in mice or telomeric damage in mouse embryonic fibroblasts (MEFs). In conclusion, our findings demonstrate that the CTED can be removed in *Mus musculus* without adverse effects. Reflecting on this, we hypothesized that the various TIN2 isoforms may share overlapping functions in maintaining telomeres to ensure cell survival and animal development.

## 2. Results

### 2.1. Characterization of Alternative Splicing and Protein Sequence Homology of TIN2 Isoforms in Humans and Mice

Previous studies have identified three distinct isoforms of human TIN2, named TIN2S, TIN2M, and TIN2L, which arise due to alternative splicing [30,31,32,33]. First, we characterized the mRNA structures of these human TIN2 isoforms. We performed sequence analysis using public data sources, including NCBI and Ensembl, to compare sequences encoding TIN2S, TIN2M, and TIN2L. Our analysis revealed alternative intron retention from exon 6 to exon 9, as shown in Figure 1A. To verify the presence of these TIN2 transcripts, we designed oligonucleotide primers targeting the amplicon from exon 5 to exon 9 for polymerase chain reaction (PCR) using cDNA generated from various human tissues. Reverse transcription PCR (RT-PCR) detected three mRNA transcripts, showing that alternative splicing events occur between exons 6 and 9 in those tissues (Figure 1B). The retained introns contain predicted in-frame stop codons, which may result in the TIN2 proteins of varying lengths corresponding to TIN2S, TIN2M, and TIN2L. We also included the *Mus musculus Tinf2* gene sequence in our analysis, which revealed that the 9-exon structure and exon sequence homology are retained between humans and mice, supporting the homology of mouse TINF2 and human TIN2L (Figure 1A).

Additionally, we performed an alignment of human and mouse TIN2 protein sequences in Clustal Omega, version 1.2.4 (Appendix A). We compared the sequences of the three human TIN2 isoforms, two distinct previously reported sequences (TIN2-1 (Q3UMZ4 in UniProt) and TIN2-2 (Q8CJ45/AF518764 in UniPort) in Figure 1C) of *Mus musculus* TINF2, and one sequence of *Mus spretus* TINF2. We found that the N-terminal 1–354 amino acid residues of all human TIN2 isoforms are identical (Appendix A). Sequence analysis revealed 63% identity between human TIN2 and mouse TINF2 for the region encoded by exons 1–6 and 42% identity for the CTED region in TIN2L (Appendix A). Furthermore, TIN2S shares 50–52% sequence identity with rodent TINF2, while TIN2L exhibits slightly higher identity (54–59%), indicating closer homology between TINF2 and human TIN2L (Appendix A). We also detected the putative CK2 phosphorylation sites, which were uncovered in a previous study [32], in TIN2L and mouse TINF2 sequences (Figure 1C), potentially indicating that TIN2L TINF2 are regulated similarly. To further explore the function of the CTED, we generated a *Mus musculus* model by deleting the CTED genomic sequence and explored the consequent cellular and physiological phenotypes.

### 2.2. Generation of a Truncated Tinf2 Allele Expressing a TIN2S-like Protein in Mice

Previous studies have indicated the functional divergence of human TIN2 isoforms in terms of how they affect telomere maintenance [32], yet their roles at the organismal level remain elusive. To investigate the potential functions of TIN2 isoforms in vivo, we opted to employ a mouse model. We sought to create a *Tinf2* allele in C57BL/6 mice that would generate a TINF2 variant protein resembling human TIN2S. Utilizing CRISPR/Cas9 technology, we introduced two double-strand breaks at intron 6 and exon 9 of the *Tinf2* gene in mouse embryos to excise the region encoding the CTED. Upon DNA repair following CRISPR-mediated cleavages, the resultant *Tinf2* allele was designed to have a stop codon immediately after exon 6 while retaining the 3′ untranslated region (UTR), thereby generating an mRNA encoding a TINF2 protein lacking the CTEDs, i.e., akin to human TIN2S. This successfully truncated *Tinf2* allele was designated the “long-form deficiency (LD)” allele (Figure 2A), and the genetic modifications could be genotyped by PCR using specifically designed primers (Figure 2B).

Upon screening 90 animals derived from CRISPR-modified embryos, we identified 14 individuals carrying *Tinf2* LD alleles. Sequence analysis of the genomic DNA from these genetically modified mice revealed 15 modified *Tinf2* LD alleles (Appendix A), with 14 of them being successfully transmitted through the germline upon crossing the founders with wildtype C57BL/6 mice. Furthermore, intercrossing heterozygous (*Tinf2^LD/+^*) mice with littermates led to the successful generation of *Tinf2^LD/LD^* animals for 12 of the 14 detected LD alleles. Subsequently, inbreeding Tinf2*^LD/LD^* mice carrying three distinct *Tinf*2 LD alleles (illustrated in Appendix A as 1, 8, and 15) produced viable offspring, underscoring the capability of *Tinf2* LD allele-carrying germ cells to undergo fertilization and embryonic development until birth. Overall, these findings indicate that the CTED of TINF2 is dispensable for fertility and embryonic survival in mice. We used mice carrying the 15th *Tinf2* LD allele for the following analysis (Appendix A).

To further validate the gene expression of the *Tinf2* LD allele, we extracted total RNA from *Tinf2^+/+^*, *Tinf2^LD/+^*, and *Tinf2^LD/LD^* MEFs and then performed RT-PCR to quantify the *Tinf2* mRNA transcript levels. Our analysis revealed that the MEFs from the *Tinf2^+/+^* and *Tinf2^LD/+^* embryos expressed mRNA transcripts corresponding to wildtype *Tinf2*, whereas short transcripts aligning with the *Tinf2* LD allele were detected in MEFs from both *Tinf2^LD/+^* and *Tinf2^LD/LD^* animals (Figure 2C).

Furthermore, we prepared protein samples from these MEFs for Western blot analysis to evaluate the expression levels of the TINF2 proteins, specifically the long isoform (wildtype TINF2) and the truncated form (TINF2LD, referred to as LD), encoded by the wildtype and the mutant *Tinf2* LD allele, respectively. As expected, the wildtype TINF2 protein was observed in both *Tinf2^+/+^* and *Tinf2^LD/+^* MEFs, whereas TINF2LD was detected in the MEFs from *Tinf2^LD/+^* and *Tinf2^LD/LD^* animals (Figure 2D). These findings support the notion that the LD allele of *Tinf2* can be transcribed into an mRNA transcript that encodes TINF2LD corresponding to human TIN2S. The resulting mutant protein contributes to animal viability and fertility, even in homozygous *Tinf2^LD/LD^* animals that exclusively express the mutant TINF2LD variant.

### 2.3. TINF2LD Protects Telomeres in MEFs

Given that TINF2LD shares similarities with human TIN2S and lacks the CTED, we used *Tinf2^+/+^*, *Tinf2^LD/+^*, and *Tinf2^LD/LD^* MEF—expressing wildtype TINF2, a mix of wildtype TINF2 and TINF2LD, or exclusively TINF2LD, respectively—to examine the role of the CTED in telomere maintenance. Initially, we investigated the telomeric localization of wildtype TINF2 and TINF2LD through immunofluorescence (IF) staining and telomeric fluorescence in situ hybridization (FISH). We observed the colocalization of TINF2 protein and telomere signals in the *Tinf2^+/+^*, *Tinf2^LD/+^*, and *Tinf2^LD/LD^* MEFs, indicating that both wildtype TINF2 and TINF2LD protein are capable of associating with telomeres (Figure 3A).

To address whether TINF2LD protein plays a role in telomere protection, we assessed the telomeric DNA damage response in MEFs according to their formation of telomere damage-induced foci (TIFs), as determined by a telomeric localization of phosphorylated histone H2AX (γH2AX). Based on cells displaying >5 TIFs (categorized as TIF+ cells), we observed that *Tinf2^+/+^*, *Tinf2^LD/+^*, and *Tinf2^LD/LD^* MEFs exhibited comparable numbers of TIFs (Figure 3B,E,F). This observation indicates that telomere damage had been maintained at a basal level upon the expression of wildtype TINF2 or TINF2LD protein in MEFs. As a control for TIF induction, we knocked down the expression of TINF2 from *Tinf2^+/+^*, *Tinf2^LD/+^*, and *Tinf2^LD/LD^* MEFs using small interfering RNAs (siRNAs) (Figure 3C,D, Appendix A), which resulted in notable and consistent increases in TIF+ cells (Figure 3B,E,F). This result indicates that the loss of TINF2, whether wildtype TINF2 or TINF2LD, promotes telomeric DNA damage. Thus, endogenous TINF2 is vital for telomere protection in *Tinf2^+/+^*, *Tinf2^LD/+^*, and *Tinf2^LD/LD^* MEFs, but deletion of the CTED does not compromise TINF2’s function in protecting telomeres.

Additionally, we assessed the ability of these cells to respond to telomere stress. Telomere stress was induced by the ectopic expression of a dominant-negative mutant of TPP1, TPP1ΔRD. TPP1ΔRD is a deletion mutant of TPP1 (TIN2 interacting protein 1, also known as ACD) that lacks the domain required for binding the protection of telomeres 1 (POT1), a deficiency known to elicit a strong telomeric DNA damage response [34,35]. We hypothesized that this treatment would provide further insight into the ability of *Tinf2^LD/LD^* MEFs to cope with impaired telomere shelterin complex formation. However, we did not observe any dramatic difference between MEF lines of different genotypes (Appendix A). Furthermore, we assessed telomere length using telomere restriction fragment (TRF) analysis (Appendix A). Both the initial telomere length (Appendix A) and telomere length across different passages (Appendix A) showed no significant differences among genotypes. Therefore, our findings support that the CTED of TINF2 is dispensable for safeguarding telomeric DNA in MEFs.

### 2.4. The CTED of TINF2 Is Not Essential for Embryonic Development or Growth of Mice

The deletion of TINF2 in mice has been shown to impede blastocyst formation and embryonic development as a consequence of telomere dysfunction [28,36]. To investigate if the presence of the *Tinf2* LD allele affects embryonic development, we examined the genotype of 180 pups resulting from the intercrossing of *Tinf2^LD/+^* mice. We observed a slight bias in favor of *Tinf2^+/+^* animals (30.6%) and reductions in *Tinf2^LD/+^* (47.8%) and *Tinf2^LD/LD^* (21.6%) animals relative to the predicted Mendelian outcomes (Appendix A). The overall prevalence of the *Tinf2* LD allele in the offspring was 45.6%. These observations indicate that the LD allele may contribute to subtle defects affecting gametes or embryonic development.

Next, we examined the physical characteristics of adult *Tinf2^+/+^*, *Tinf2^LD/+^*, and *Tinf2^LD/LD^* mice (Figure 4). More specifically, we analyzed the brain, spleen, and kidney of 4-month-old animals for potential abnormalities. However, we detected no significant differences in either the general appearance or specific organ structures of the animals (Figure 4A,B). The body weight analysis revealed that the *Tinf2^LD/+^* and *Tinf2^LD/LD^* mice grew at a similar pace postnatally as *Tinf2^+/+^* animals (Figure 4C,D), with only marginal average weight differences among male animals (1% to 8.15% of average body weight differences for recordings from 4 to 14 weeks old) observed after the experiment. Similarly, the survival curves across genotypes indicated comparable lifespans (Figure 4E). These results reinforce the notion that the LD allele is capable of supporting the normal development and growth of adult mice.

Although our adult *Tinf2^+/+^*, *Tinf2^LD/+^*, and *Tinf2^LD/LD^* mice exhibited no obvious phenotypic differences, we considered the possibility that such differences might ultimately emerge over successive generations. To explore that possibility, we initiated inbreeding among *Tinf2^+/+^* or *Tinf2^LD/LD^* animals to produce second-generation (G2) offspring. This breeding process extended over two years and produced successive generations, reaching the seventh generation (G7) for both *Tinf2^+/+^* and *Tinf2^LD/LD^* mice (Appendix A). Moreover, the *Tinf2^LD/LD^* mice exhibited a comparable reproductive ability, producing both male and female offspring at similar rates (Appendix A). Observations of these animals to date have revealed that the G7 generations of both the *Tinf2^+/+^* and *Tinf2^LD/LD^* mice remained fertile and presented no physical abnormalities, implying an absence of cumulative effects for late-generation *Tinf2^LD/LD^* mice despite the sole expression of the mutant protein TINF2LD. Together, our collective findings indicate that, despite lacking the CTED, TINF2LD is sufficient for cellular telomere maintenance and can thus support the normal development and growth of laboratory mice.

## 3. Discussion

TIN2 is essential for telomere protection and animal viability. The functional characterization of human TIN2 isoforms using cultured cells has revealed their involvement not only in telomere protection but also in telomerase regulation [32]. TIN2 isoforms differ in the CTED, which may dictate isoform-specific telomeric functions. Consistently, it has been shown that the phosphorylation of TIN2L at the CTED modulates the interaction between TIN2 and TRF2, promoting telomere lengthening [32]. The common laboratory mouse, *M. musculus*, expresses TINF2 hosting the CTED and most closely resembles human TIN2L based on similarities in their protein sequences (Figure 1C, Appendix A). Unlike the ubiquitous expression of TIN2 isoforms in human tissues (Figure 1B), our analysis of MEFs (Figure 2C) and embryonic stem cells did not detect the expression of TINF2 isoforms, indicating that mouse TINF2 does not undergo alternative splicing. However, it is noteworthy that in the wild species of mouse, *Mus spretus*, alternative splicing generates both short and long TINF2 isoforms [31], analogous to humans. Moreover, it is known that *M. spretus* possesses human-sized telomeres, which are relatively shorter than those of *M. musculus*, suggesting that TINF2 isoforms may differentially regulate telomere dynamics in rodents. Thus, evolutionary divergence in the alternative splicing that regulates the expression of TIN2 isoforms may shape the mechanism of telomere maintenance in vertebrates [37].

To understand the function of TIN2 isoforms in vivo, we generated genetically modified mice expressing a truncated TINF2 protein, TINF2LD, which lacks the CTED and is analogous to human TIN2S (Figure 2A). Unlike the embryonic lethality caused by the deletion of TINF2, the *Tinf2^LD/LD^* mice were viable, indicating that the CTED of TINF2 is not essential for embryonic development or animal viability in *M. musculus*. In addition, we observed that TINF2LD proteins localized to telomeres in the MEFs of *Tinf2^LD/LD^* mice (Figure 3A) and did not induce a telomere DNA damage response (Figure 3B,E,F), supporting that TINF2LD protein is proficient in telomere protection. Moreover, adult *Tinf2^LD/LD^* mice exhibited no overt phenotypic abnormalities (Figure 4), indicating that the CTEDs are dispensable for normal growth and fecundity in *M. musculus*. To investigate the lack of phenotypic changes in mice, we analyzed the putative casein kinase 2 (CK2) phosphorylation site [32], represented by the evolutionarily conserved amino acid sequence TIGDLVLDSDEEE, which is coded by the exon 8 of mouse *Tinf2* (Figure 1C). However, a telomeric function for CK2 has not been described for mice, though it has been for *Schizosaccharomyces pombe* (*S. pombe*) [38,39] and humans [40,41], indicating potential interspecies differences in how CK2 contributes to regulating mouse TINF2-mediated telomeric protection. Thus, our findings reveal ambiguity regarding the CK2-dependent phosphorylation of the CTED in mouse TINF2. This ambiguity could result from either an absence of phosphorylation or a lack of its functional contribution to phenotypic changes. Further investigations may be required to fully elucidate CK2-dependent phosphorylation and its role in regulating mouse TINF2 proteins.

Nevertheless, we did detect subtle differences in inheritance patterns (Appendix A) and male mouse body weight (Figure 4C) for *Tinf2^LD/LD^* mice. It has been demonstrated previously that the human TIN2S and TIN2L isoforms may exert distinct functions in regulating telomere length, particularly due to the specific phosphorylation of the CTED of TIN2L [32]. Therefore, it is possible that the deletion of the CTED in mice could disrupt telomere maintenance in certain tissues, leading to minor organism-level effects. Additionally, it would be interesting to investigate if these effects are related to the non-telomeric functions of TINF2, such as its reported localization in mitochondria [42]. Furthermore, the modest phenotypic variations between wildtype and CTED-deficient mice suggest the need to elucidate the full spectrum of TINF2 functions under stress conditions. Overall, our findings reveal that TIN2 isoforms may exert redundant in vivo functions in terms of telomere protection and maintenance, which support animal development and growth. Future comprehensive studies using our TINF2 mouse model could elucidate the intricacies of telomere biology across species. Additionally, our animal models can be used in studying the unique nature of alternative multiple intron retention events. Additional research is warranted to determine the translational relevance of our findings to humans.

## 4. Materials and Methods

### 4.1. Generation of the Tinf2 LD Allele in Mouse

The *Tinf2* LD mice were generated in-house by the Transgenic Core Facility at the Institute of Molecular Biology, Academia Sinica, Taiwan, utilizing a CRISPR/Cas9-mediated approach on zygotes of C57BL/6 mice with two guide RNAs targeting the *Tinf2* sequence: guide sequence 1 (5′-AGGCAGGGAAAGTATTCCTAAGG-3′) and guide sequence 2 (5′-CGGGATGTAGTCACAAAACATGG-3′). Targeted zygotes were transferred to surrogate mothers for embryonic development into mature animals. Donor mice providing zygotes were produced in-house by the Transgenic Core Facility, and surrogate mice were obtained from BioLASCO Taiwan Co., Ltd. (Taipei, Taiwan, R.O.C.) Mouse tail genomic DNA was screened using PCR with primers (5′-AGCCAGTCTGCACGGAGGAGG-3′ and 5′-GCAGAGCTGCAGCAGAAGATGG-3′), resulting in products of 869 base pairs (bps) for the wildtype *Tinf2* allele and 264 bps for the modified *Tinf2* LD allele. DNA sequencing was used to confirm the absence of frameshift mutations. The mapped sequencing results are shown in Appendix A. The animals whose genomic DNA passed the screening became founders for further generational offspring. A total of 14 animals were chosen as the founders, and 15 distinct mutated alleles were identified. The 15th allele was expanded for cellular and adult phenotype analyses. Animal experiments were conducted with permission from the Institutional Animal Care and Use Committee (protocol ID: IACUC-17-11-1132) and in strict accordance with its guidelines, as well as those outlined in the Council of Agriculture Guidebook for the Care and Use of Laboratory Animals in Taiwan.

### 4.2. Reverse Transcription Polymerase Chain Reaction (RT-PCR)

Primary MEF lines were lysed to extract total RNA using an RNA Mini Kit (Amersham, Chalfont Saint Giles, UK). Reverse transcription was performed with iScript Reverse Transcriptase Supermix (Bio-Rad, Hercules, CA, USA) to generate cDNA for further analyses. PCR primers (5′-AGCCAGTCTGCACGGAGGAGG-3′ and 5′-GCAGAGCTGCAGCAGAAGATGG-3′, the same pair of primers that were used for genotyping with genomic DNA) were used to determine *Tinf2* alleles in MEFs. The PCR generated products of 421 bps for the wildtype *Tinf2* allele and 264 bps for the modified *Tinf2* allele.

### 4.3. Generation of Mouse Embryonic Fibroblasts (MEFs)

Pregnant mice at 13 or 14 d.p.c. (day post-coitum) were euthanized by cervical dislocation before dissecting out the uterine horns. Pregnant animals were used to generate independent MEF lines. Individual embryos were dissected from the uterine horns, removing the head and internal organs, and then subjected to trypsinization and pipetting to generate single-cell suspensions. Isolated cells were cultured in MEF medium [comprising Dulbecco’s modified eagle’s medium (DMEM) supplemented with 10% fetal bovine serum (FBS), 0.5X penicillin/streptomycin, and 0.1 mM β-mercaptoethanol] under conditions of 37 °C with 5% CO_2_ and 3% O_2_. Primary MEF lines were cryopreserved at −80 °C after the first passage. Cellular analyses were performed after thawing, with all experiments conducted within 6–10 passages.

### 4.4. Protein Extraction and Western Blotting

MEF cell extracts were obtained using 1X SDS sample buffer (62.5 mM Tris pH 6.8, 10% glycerol, 2% SDS, 0.01% bromophenol blue, and 10% β-mercaptoethanol), followed by denaturation by heating to 90 °C for 10 min. Denatured protein samples were loaded onto SDS-PAGE gels, separated, and transferred to nitrocellulose membranes. Blocking was performed in phosphate-buffered saline (PBS) containing 5% milk and 0.1% Tween-20. Primary antibodies targeting mouse TINF2 were generated by immunizing rabbits using bacterial-expressed and purified His-tagged mouse TINF2 (1–339) protein. The anti-TINF2 antibodies and horseradish peroxidase (HRP)-conjugated secondary antibodies were used at dilutions of 1:1000 and 1:2000, respectively. HRP-labeled ACTB antibodies (HRP-60008, ProteinTech, Rosemont, IL, USA) were used at a dilution of 1:5000 as an internal control. Signal detection was accomplished using either enhanced chemiluminescence (ECL) or SuperSignal West Femto Maximum Sensitivity reagents (Thermo Fisher Scientific, Waltham, MA, USA).

### 4.5. Immunofluorescence and Fluorescence In Situ Hybridization (IF-FISH)

Cells were cultured on coverslips, permeabilized, fixed, and treated with primary antibodies against customized generated mouse TINF2 antibody or γH2AX (05-636, Millipore, Burlington, MA, USA), followed by secondary antibodies (mouse IgG Alexa Fluor 488, A-11001, Invitrogen, Waltham, MA, USA). After secondary antibody treatment, cells were fixed again, dehydrated, and incubated with a probe mixture (70% formamide, 0.5% blocking reagent (Roche, Basel, Switzerland), 250 nM TMR-conjugated (CCCTAA)_3_-PNA probe, 10 mM Tris pH 7.4). The coverslips were counterstained with DAPI, dehydrated, and mounted for imaging using a DeltaVision Deconvolution microscope (Dallas, TX, USA). The colocalization events were quantified by counting nuclei on the coverslips.

### 4.6. Analyses of Animal Survival, Body Weight, and Organ Structures

Animals were housed under specific pathogen-free (SPF) conditions. The breeding of *Tinf2^LD^* animals involved mating male and female animals aged 3 to 9 months. The genotyping of newborn animals was performed by PCR using *Tinf2* primer pairs (5′-AGCCAGTCTGCACGGAGGAGG-3′ and 5′-AGGCAACTCAAGTCTGGACTTCC-3′). Body weight was recorded without further treatment, and natural deaths were noted. The statistical analysis of body weight utilized a modified Chi-square analysis [43], whereas differences in animal mortality between genotypes were assessed using a log-rank test. To obtain animal organs for structural examinations, the littermates of 3 mice were euthanized using carbon dioxide before the surgery.

### 4.7. Stable Expression in Cells with Viral Infection

For the induction of telomere stress, cells were engineered for the stable expression of target constructs. Expression constructs coding TPP1-YFP or TPP1ΔRD were transfected into 293T cells using Lipofectamine 3000 (Thermo Fisher, Waltham, MA, USA) along with pcl-Ampho constructs to generate virus-containing medium over three days. The collected virus-containing medium was diluted at a 1:5 ratio with MEF medium and used to infect experimental MEF lines. Infected cells were seeded onto coverslips for IF-FISH analysis.

### 4.8. Telomere Restriction Fragment Analysis

Genomic DNA was extracted from MEF lines at the indicated passage. DNA samples were digested with RsaI and HinfI restriction enzymes (NEB) for 16 h in 37 °C. The digested DNA fragments were separated via electrophoresis under pulse-field conditions. Following electrophoresis, the gel was dried and hybridized with α-P^32^-dCTP-labeled TTAGGG probes to detect telomere sequence-containing fragments. Radioactive signals were visualized using a Typhoon FLA 9500 fluorescence image analyzer (GE Healthcare, Chicago, IL, USA).

### 4.9. Statistics

Unless otherwise specified, statistical analyses were performed using GraphPad Prism, version 9.4.1. A one-way ANOVA was used for multiple comparisons, followed by Tukey’s post hoc test to assess differences between the two groups.

## Figures and Tables

**Figure 1 ijms-26-02414-f001:**
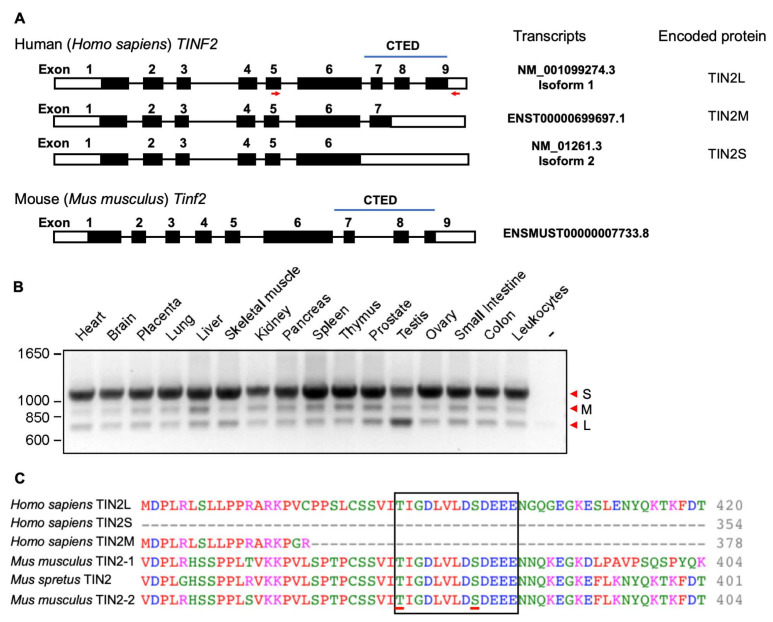
Characterization of human *TIN2* and mouse *Tinf2* mRNA expression. (**A**) Schematic of the structural organization of human *TIN2* (hTIN2) and mouse *Tinf2* (mTINF2) genomic sequences. Exons are represented as numbered rectangular shapes, and connecting lines indicate introns splicing between exons. Open rectangular shapes represent the untranslated region (UTR), while closed rectangular shapes (black rectangles) indicate the coding sequences in exons. Sequences encoding the CTED of human TIN2L and mouse TINF2 are marked. Red arrows indicate the genomic sequence used as a primer for the PCR-based amplification shown in (**B**). (**B**) Detection of hTIN2 isoform expression in adult tissues. cDNAs from various human adult tissues were used for PCR primers indicated in (**A**). The arrowheads labeled S, M, L represent the detected signals corresponding to TIN2S, TIN2M, and TIN2L, respectively. (**C**) Human and mouse TIN2 protein sequence alignment results in the regions corresponding to the putative CK2 phosphorylation sites. The putative binding site is highlighted in the open rectangular shapes, and the potential phosphorylation sites are underlined. Full-length sequence alignment results are provided in Appendix A. Full-length gels and blots are provided in Appendix A.

**Figure 2 ijms-26-02414-f002:**
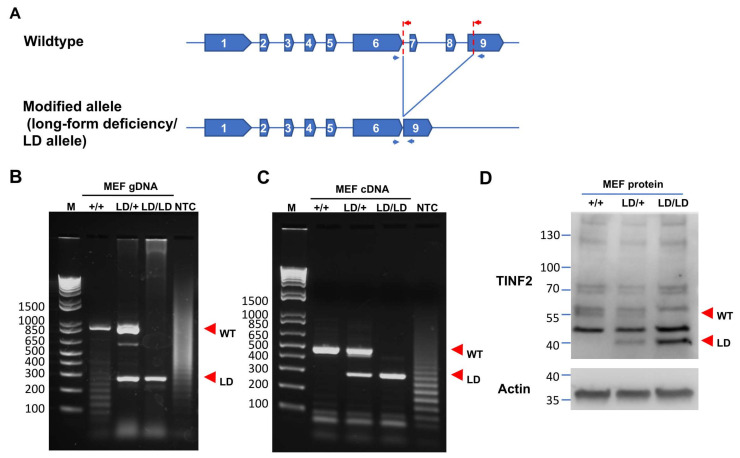
Generation of the *Tinf2* long-form deficiency (LD) allele in mouse. (**A**) Schematics illustrating the gene structure of the wildtype (WT) and long-form deficiency (LD) alleles. Two guide RNAs (red arrows), flanking mouse *Tinf2* intron 6 and partial exon 9, along with Cas9-expressing constructs, were designed to generate a modified *Tinf2* allele in mice, replicating human TIN2S. Blue arrows denote PCR genotyping primers. (**B**) Representative genotyping results of mouse embryonic fibroblasts (MEFs). MEFs were isolated from embryos on embryonic day 14.5 (E14.5) for genomic DNA extraction. PCRs were carried out with the primers indicated in (**A**), and the resulting oligonucleotides were subjected to agarose gel electrophoresis. (**C**) Representative RT-PCR results obtained from MEFs. mRNA samples were extracted from the indicated MEFs and subjected to reverse transcription to generate cDNA samples for PCRs with the primers indicated in (**A**). PCR products were analyzed by agarose gel electrophoresis. (**D**) Western blotting for TINF2 and actin protein expression in MEFs of different genotypes. Red arrowheads highlight bands corresponding to the wildtype (WT) or LD allele. Full-length gels and blots are provided in Appendix A.

**Figure 3 ijms-26-02414-f003:**
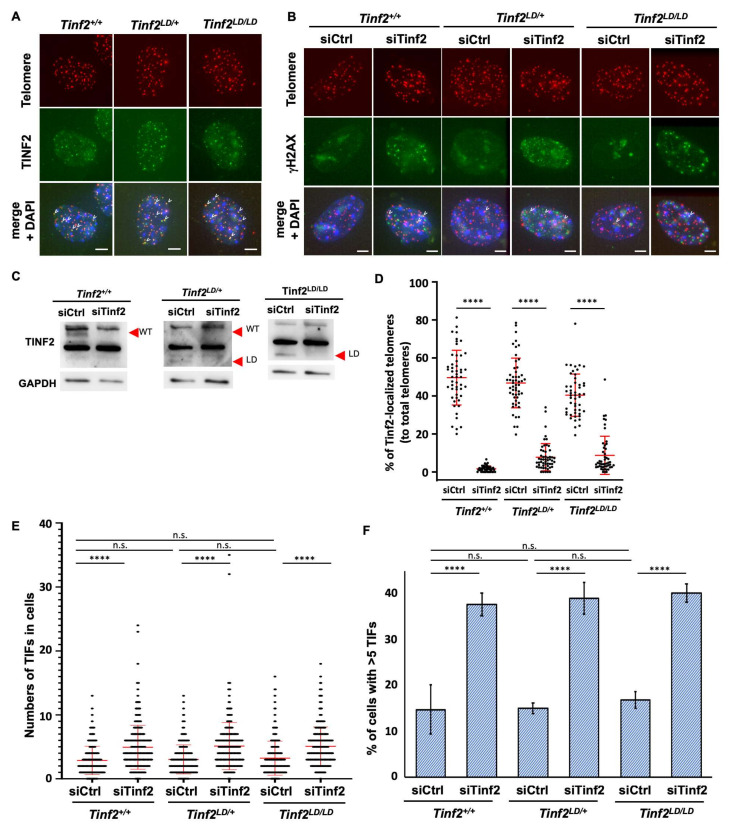
Telomeric localization and protection function of wildtype TINF2 and TINF2LD in MEFs. (**A**) Representative images of IF-FISH illustrating expression of mouse TINF2 protein and telomere signal in MEFs. Telomere signals were visualized using a telomere probe (red), whereas endogenous mouse TINF2 signals were detected using mouse anti-Tinf2 antibodies (green). Representative telomere-localized mouse TINF2 (colocalized signals) is indicated by white arrowheads. Scale bar: 2 μm. (**B**) Representative images of control and siRNA-treated MEFs of different *Tinf2* genotypes (*Tinf2^+/+^*, *Tinf2^LD/+^*, and *Tinf2^LD/LD^*). White arrows indicate colocalized signals of γH2AX and telomere puncta, indicative of telomere damage-induced foci (TIFs). Cells displaying more than five TIFs were defined as TIF+ cells. Scale bar: 2 μm. (**C**) Western blotting analysis for TINF2 and GAPDH in siRNA-treated primary MEFs with different genotypes. WT bands correspond to wild-type protein, while LD bands represent TINF2LD. (**D**) Percentage of telomeres with mouse TINF2 signals for different genotypes with *Tinf2* or control knockdown treatment. Data are presented as mean ± SD. More than 50 cells were counted for each condition. Statistical analysis was conducted using one-way ANOVA. ****, *p*-value < 0.0001. Representative images are shown in Appendix A. (**E**) Numbers of TIFs in each cell. Data represent three primary MEF lines, with over 300 cells assessed in each group. Data are presented as mean ± SD, with longer lines denoting the mean value and shorter lines representing standard deviation. Statistical analysis was conducted using one-way ANOVA. n.s., not significant; ****, *p*-value < 0.0001. (**F**) Percentages of TIF+ cells for various genotypes. For each genotype, analysis was performed on three independent primary MEF lines, with the percentage of TIF+ cells determined from more than 100 cells of each line. Data are presented as mean ± SD. Statistical analysis was conducted using one-way ANOVA. n.s., not significant; ****, *p*-value < 0.0001. Further statistical details are shown in Appendix A.

**Figure 4 ijms-26-02414-f004:**
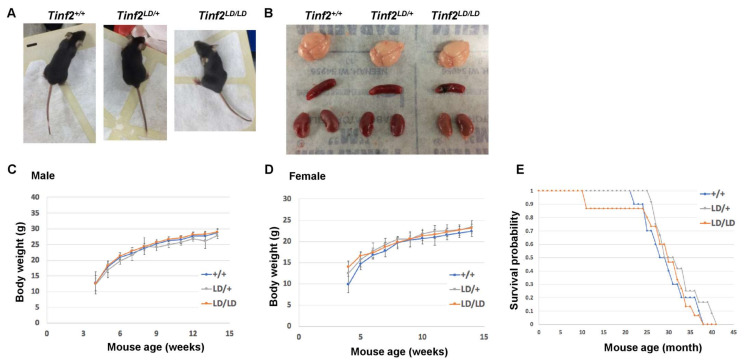
Phenotypic comparison of adult animals of different genotypes. (**A**) Gross views of mice representing different genotypes. (**B**) Gross views of mouse brain, spleen, and kidney. (**C**,**D**) Body weight trajectories over time (4–14 weeks) of male and female mice. For the male body weight, the analysis comprised 4 *Tinf2^+/+^*, 6 *Tinf2^LD/+^*, and 5 *Tinf2^LD/LD^* animals. For the female body weight, the analysis comprised 3 *Tinf2^+/+^*, 4 *Tinf2^LD/+^*, and 6 *Tinf2^LD/LD^* animals. Data are represented as mean ± SD. A Chi-square-based statistical analysis was conducted for male body weight: *Tinf2^+/+^* vs. *Tinf2^LD/+^*: *p*-value = 0.19; *Tinf2^+/+^* vs. *Tinf2^LD/LD^*: *p*-value = 0.94; and *Tinf2^LD/+^* vs. *Tinf2^LD/LD^*: *p*-value = 0.0006. The same statistical analysis was conducted on the data of female body weight: *Tinf2^+/+^* vs. *Tinf2^LD/+^*: *p*-value = 0.45; *Tinf2^+/+^* vs. *Tinf2^LD/LD^*: *p*-value = 0.91; and *Tinf2^LD/+^* vs. *Tinf2^LD/LD^*: *p*-value = 0.20. (**E**) Survival curves for mice of the indicated genotypes were tracked over a period of up to 4 years. The analysis comprised 10 *Tinf2^+/+^*, 12 *Tinf2^LD/+^*, and 15 *Tinf2^LD/LD^* animals. Survival probability is depicted as a Kaplan–Meier plot. An overall log-rank test revealed an χ^2^ value of 2.02 and a *p*-value of 0.64 [for independent comparisons: *Tinf2^+/+^* vs. *Tinf2^LD/+^*: χ^2^ = 1.22, *p*-value = 0.73; *Tinf2^+/+^* vs. *Tinf2^LD/LD^*: χ^2^ = 0.17, *p*-value = 0.32; and *Tinf2^LD/+^* vs. *Tinf2^LD/LD^*: χ^2^ = 1.36, *p*-value = 0.76].

## Data Availability

The original contributions presented in this study are included in the article/Appendix A. Further inquiries can be directed to the corresponding author.

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
