# Peer review of "C-Terminal Extended Domain-Independent Telomere Maintenance: Modeling the Function of TIN2 Isoforms in *Mus musculus"

_ijms, 2025, doi:10.3390/ijms26062414_

Round 1
Reviewer 1 Report
Comments and Suggestions for Authors
The authors employed gene editing, IF-FISH, and Western blotting to investigate the role of TIN2 and its C-terminal extended domain (CTED) in telomere maintenance in both mice and mouse embryonic fibroblasts (MEFs). These experiments aimed to mimic the human TIN2S and TIN2L isoforms. The results demonstrated that TINF2 is essential for telomere protection in Tinf2+/+, Tinf2LD/+, and Tinf2LD/LD MEFs, but deletion of the CTED did not compromise TINF2’s function in safeguarding telomeres.
I have carefully reviewed your manuscript and found the experimental design to be sound, the workload substantial, and the writing style to be clear and concise. However, I have a few minor questions for the authors to address:
1. Given the existence of multiple TIN2 isoforms in mice (NCBI), do all of them possess the C-terminal extended domain (CTED)?
2. Are there any copy number variations in the mouse TIN2 gene?
3. As you have stated that the CTED plays a unique role in telomere maintenance, why do the gene-edited mice not exhibit any abnormalities?
4. Why was a chi-squared test not performed on Table S2?
Author Response
Comments 1: Given the existence of multiple TIN2 isoforms in mice (NCBI), do all of them possess the C-terminal extended domain (CTED)?
Response 1: Based on the NCBI database, the mouse TINF2 gene gives rise to seven transcripts. All of these transcripts include the C-terminal extended region, though alternative splicing may lead to distinct isoforms. In our genetically modified mouse model, we specifically deleted the common region that is expected to be functionally relevant.
Comments 2: Are there any copy number variations in the mouse TIN2 gene?
Response 2: To our knowledge, there is no evidence suggesting that the TINF2 gene in the mouse genome exists in multiple copies.
Comments 3: As you have stated that the CTED plays a unique role in telomere maintenance, why do the gene-edited mice not exhibit any abnormalities?
Response 3: We hypothesize that the functional importance of CTED in mouse TIN2 may be mitigated by species-specific regulatory mechanisms. Additionally, phosphorylation of the CTED may differ between mice and humans, as discussed in Lines 330-339 of our manuscript. Our phenotypic analysis showed no significant defects in mouse development or telomere damage response, but we acknowledge that subtle or tissue-specific abnormalities might exist. We discussed on potential compensatory mechanisms and future research directions in Lines 346-352.
Comments 4: Why was a chi-squared test not performed on Table S2?
Response 4: We appreciate the reviewer’s suggestion and have now included a chi-square test for Hardy-Weinberg equilibrium in the revised figure legend for Figure S2.

Reviewer 2 Report
Comments and Suggestions for Authors
Use mouse genetics, the authors studied whether the C-terminal domain of TINF2 (TIN2 protein in mouse) has any potential functional roles in regulating telomere and therefore the growth and health of cells in mice. Their results suggest that the deletion of CTED in mice show no overall obvious defective phenotypes in the animal, as well as in MEF cells in culture, adding new information to the understanding of telomere regulating protein TIN2. The data also indicate the differences in the regulatory modes of telomere between human and mice. This work should be of interest to researchers in the field. However, the manuscript contains some major flaws that impede its significance and should be corrected or improved before being considered for publication. Main concerns are:
1) The nomenclature of mouse TINF2 protein and gene are not described clearly, sometimes with the authors’ own definition, these could cause confusions and make it difficult to understand. One suggestion is that to use consistent naming for the genes and proteins without creating one, e.g. for the homology analysis: what are the differences/similarities between mouse TIN2-1 and TIN2-2 (C-terminal aa sequences are quite different, full length being 414-aa long, CCDS27125.1?), are they encoded by the same gene (allele)? name “long-form deficiency” is confusing, because no functional defects were observed, TINF2-S might be more appropriate, however, it is an artificial mutant, not an natural isoform as in human. Lines 182-190, renaming TINF2 to -L and -S is quite confusing, suggest to use the same name “TINF2” and its mutant form;
2) Some overstatements in the main text, e.g. the conclusion in line 174-175 seems premature without any data showing the reproduction and survival (liter size, mating, overall health, etc.), comparing to the wild type, although growth analyses were presented later. The CK2 phosphorylation of TINF2 is hypothetical, can the phosphorylation of TINF2 be examined in mice?
Other points:
1) Line 65, 73, human TIN2L interacts with nuclear matrix, lacking CTED induced dysregulated telomere length in RPE1-hTERT cells, indicating the functional roles of CTED: TIN2M and TIN2S don’t compensate for TIN2L. How the data in mouse would be interpreted regarding this? Mouse express only TINF2, no other isoforms: line 84-84: isoforms have overlapping functions?
2) Figure 1: indicate the stop codons caused by “retained introns” in the diagram; 1B labels S, M, and L from top to bottom, are these using the pair of primers shown in 1A?
3) Lines 171-175, specify what type of mutations from the founder mice were used for establishing the mutant mouse line that were further use for subsequent analyses; show genotyped mutant sequences vs. wild type;
4) Figure 2A: label the blue arrows’ position correctly; 2C, which part of the cDNA was used as the indicator for +/+, and -/-? Are the primers here the same as the genotyping primers? 2D, indicate the molecular weight markers and WT bands?
5) Figure 3, describe clearly why use siRNA assay, what method used to induce DNA damage, in the main text; the protein bands are very un-clear in 3C; show corresponding fluorescent images for the quantitation in 3D; lines 231-232: siRNA is knock down assay, not deletion of proteins;
6) Lines 275-286, presenting some recorded fertility data (liter numbers and sizes, male and females through G7) may support the conclusion;
7) Lines 107-119, 151-164, 202-221, 260-274mixed figure legend with the main text;
8) Only MEF cells were used to examine the roles of CTED, would other cell types be different in the same assay? Could telomere length be checked over the growth period of cells?
9) More details should be provided for Materials and Methods.
Author Response
Comments 1: The nomenclature of mouse TINF2 protein and gene are not described clearly, sometimes with the authors’ own definition, these could cause confusions and make it difficult to understand. One suggestion is that to use consistent naming for the genes and proteins without creating one, e.g. for the homology analysis: what are the differences/similarities between mouse TIN2-1 and TIN2-2 (C-terminal aa sequences are quite different, full length being 414-aa long, CCDS27125.1?), are they encoded by the same gene (allele)? name “long-form deficiency” is confusing, because no functional defects were observed, TINF2-S might be more appropriate, however, it is an artificial mutant, not an natural isoform as in human. Lines 182-190, renaming TINF2 to -L and -S is quite confusing, suggest to use the same name “TINF2” and its mutant form;
Response 1: We appreciate the reviewer’s concern regarding terminology. Our bioinformatics analysis identified two candidate protein sequences for mouse TIN2—Q8CJ45/AF518764 and Q3UMZ4—which differ in their C-terminal regions. For clarity, we have designated these as TIN2-1 (Q3UMZ4) and TIN2-2 (Q8CJ45/AF518764) in our manuscript (Lines 123-124).
Regarding the term “long-form deficiency” (LD): This refers to the specific allele we generated, which lacks the ability to produce full-length TIN2L-like transcripts and proteins. We have clarified this in the manuscript (Line 148-150). Furthermore, to avoid confusion, we will refer to the protein product of the Tinf2LD allele as TINF2LD throughout the manuscript.
Comments 2: Some overstatements in the main text, e.g. the conclusion in line 174-175 seems premature without any data showing the reproduction and survival (liter size, mating, overall health, etc.), comparing to the wild type, although growth analyses were presented later. The CK2 phosphorylation of TINF2 is hypothetical, can the phosphorylation of TINF2 be examined in mice?
Response 2: We agree that our conclusions should be more cautiously stated. We have revised Lines 172-177 to make it clear that we did not observe major reproductive or survival defects in our mouse model, but further studies are needed to rule out subtle phenotypic effects.
Regarding to CK2 phosphorylation of TINF2, our study does not demonstrate significant functional differences between TINF2 variants, and therefore, whether wild-type TINF2 undergoes CK2 phosphorylation does not impact our conclusions. While CK2 phosphorylation may play a role in regulating TINF2 function, investigating this mechanism is beyond the scope of the present study.
Comments 3: Line 65, 73, human TIN2L interacts with nuclear matrix, lacking CTED induced dysregulated telomere length in RPE1-hTERT cells, indicating the functional roles of CTED: TIN2M and TIN2S don’t compensate for TIN2L. How the data in mouse would be interpreted regarding this? Mouse express only TINF2, no other isoforms: line 84-84: isoforms have overlapping functions?
Response 3: After carefully reviewing reference 26, we note that the truncation mutant used in this study lacks both the C-terminal TRF1 binding site and CTED. This suggests that the observed phenotypic outcomes could result from loss of either domain. To prevent potential misunderstandings, we have revised the description in Line 73-74 of the main text for greater accuracy.
In response to the reviewer’s question, since mouse TINF2 naturally contains CTED, we propose that wild-type mouse TINF2 functions analogously to human TIN2L. Therefore, our genetic modification, which creates a new allele termed the “LD allele of mouse TINF2,” serves as a functional equivalent of human TIN2S. This model allows us to distinguish between the roles of TIN2L and TIN2S in mice.
Additionally, we observed functional redundancy between wild-type TINF2 and the LD allele, supporting our initial hypothesis that TIN2S and TIN2L may share overlapping roles. To clarify this point, we will revise Lines 85-87 to explicitly explain our findings on functional redundancy between the isoforms.
Comments 4: Figure 1: indicate the stop codons caused by “retained introns” in the diagram; 1B labels S, M, and L from top to bottom, are these using the pair of primers shown in 1A?
Response 4: We have indicated the stop codons caused by retained introns in the diagram and confirmed that Figure 1B PCR results correspond to the primers shown in Figure 1A. To clarify the description, we revised the sentences in Line 113-114.
Comments 5: Lines 171-175, specify what type of mutations from the founder mice were used for establishing the mutant mouse line that were further use for subsequent analyses; show genotyped mutant sequences vs. wild type;
Response 5: We appreciated for the advices. We will specify the mutant alleles used in line 172-177 and the Material and Method section (Line 374-376).
Comments 6: Figure 2A: label the blue arrows’ position correctly; 2C, which part of the cDNA was used as the indicator for +/+, and -/-? Are the primers here the same as the genotyping primers? 2D, indicate the molecular weight markers and WT bands?
Response 6: We have corrected the placement of blue arrows in Figure 2A. Additionally, we have indicated molecular weight markers in Figure 2D. In figure 2C, the samples were selected by the genotyping results, and the cDNA samples were generated from the reverse transcription of RNA samples from MEF cells.
Comments 7: Figure 3, describe clearly why use siRNA assay, what method used to induce DNA damage, in the main text; the protein bands are very un-clear in 3C; show corresponding fluorescent images for the quantitation in 3D; lines 231-232: siRNA is knock down assay, not deletion of proteins;
Response 7: We now describe the rationale for using siRNA knockdown in Line 232 and have included fluorescent images in Figure S3.
Comments 8: Lines 275-286, presenting some recorded fertility data (liter numbers and sizes, male and females through G7) may support the conclusion;
Response 8: We have added recorded fertility data (litter size, sex ratios) to Supplementary Table S4 and referenced it in Lines 291-295.
Comments 9: Lines 107-119, 151-164, 202-221, 260-274mixed figure legend with the main text;
Response 9: We appreciated for the notion of this formatting issue. We have corrected the format figure legends (Line 108-120, 152-165, 204-224, 274-287)
Comments 10: Only MEF cells were used to examine the roles of CTED, would other cell types be different in the same assay? Could telomere length be checked over the growth period of cells?
Response 10: We acknowledge that other cell types might exhibit distinct phenotypes, but our study focused on MEFs due to their well-established use in telomere biology research. Future studies will explore other tissues.
Regarding telomere length, our new analysis (now included in Supplementary Fig. S5) shows that telomere length remains stable across multiple passages. However, given the presence of telomerase in mouse cells, we recognize the need for further investigation, possibly by crossing our mouse model with telomerase-deficient mice. The telomere length phenotype is described in Lines 247-250.
Comments 11: More details should be provided for Materials and Methods.
Response 11: We have expanded the Materials and Methods (Line 372-376, 382, 385-386, 391-392, 397-399, 415, and additional sections in Line 433-451) section to provide additional methodological details.

Reviewer 3 Report
Comments and Suggestions for Authors
The study investigates the role of TIN2 isoforms in Mus musculus, focusing on CTED-independent telomere maintenance. While the manuscript presents interesting findings on telomere biology and TIN2 isoform functions, several critical aspects require clarification and additional experimentation. Below are my comments and suggestions for improvement.
Major revision:
- The authors claim that TIN2S is sufficient for telomere protection, but evidence is primarily based on TIF formation assays. To strengthen the conclusions, I recommend:
- Telomere length analysis (e.g., TRF analysis or qPCR) in wild-type vs. TINF2S-only MEFs over multiple passages.
- Telomerase activity assay to determine if TINF2S affects telomerase recruitment.
- While the study suggests that CTED is dispensable, the authors should explore its role under stress conditions (e.g., oxidative stress or replication stress). Treat cells with replication stress-inducing agents (aphidicolin or hydroxyurea) and evaluate telomere integrity in TINF2S vs. TINF2L-expressing MEFs.
- The study lacks direct evidence showing whether TIN2S retains interactions with shelterin components at a comparable level to TIN2L. Perform co-immunoprecipitation (Co-IP) and proximity ligation assays to assess TIN2S binding efficiency with TRF1, TRF2, and TPP1.
- Since telomere dysfunction often manifests over time, long-term phenotypic effects should be assessed. Monitor telomere shortening and cellular senescence markers in aged TINF2S-expressing mice.
Minor Concerns:
- Some Western blot images (Figure 2D) appear low resolution and need better quantification (e.g., densitometry analysis).
- Statistical analysis should include effect size and confidence intervals, not just p-values.
- The manuscript states that alternative splicing is absent in Mus musculus but present in Mus spretus. Given the evolutionary importance, the authors should discuss whether forced alternative splicing of TINF2 in Mus musculus alters telomere dynamics.
- Some critical studies on TIN2 isoform function in humans (e.g., de Lange, 2018) are missing and should be cited.
Author Response
Comments 1: The authors claim that TIN2S is sufficient for telomere protection, but evidence is primarily based on TIF formation assays. To strengthen the conclusions, I recommend:
Telomere length analysis (e.g., TRF analysis or qPCR) in wild-type vs. TINF2S-only MEFs over multiple passages.
Telomerase activity assay to determine if TINF2S affects telomerase recruitment.
Response 1: We have provided TRF assay data from three independent MEF lines each for Tinf2+/+, Tinf2LD/+, and Tinf2LD/LD genotypes in Figure S5A, and data from representative MEF lines cultured over 2-8 passages in Figure S5B. The results show no significant variation in telomere length among the three genotypes, nor over the culture period. This suggests that the genetically modified TINF2 protein, TINF2LD protein can support telomere length homeostasis in MEFs and presumably exclude the likelihood that the TINF2LD protein affects telomerase recruitment (the 2nd point). To our knowledge, the telomerase activity assay (TRAP assay) is an in vitro method, and therefore does not provide information on in vivo telomerase recruitment.
Comments 2: While the study suggests that CTED is dispensable, the authors should explore its role under stress conditions (e.g., oxidative stress or replication stress). Treat cells with replication stress-inducing agents (aphidicolin or hydroxyurea) and evaluate telomere integrity in TINF2S vs. TINF2L-expressing MEFs.
Response 2: We have conducted preliminary experiments using dominant-negative TPP1 expression to induce telomere stress. These results, which show telomere damage responses through TIF formation, have been incorporated into Supplementary Fig. S4 and described in Lines 239-247.
Additionally, this study focuses on understanding the role of TINF2 variants in telomere maintenance under physiological conditions, where human TIN2 isoforms exhibit distinct activities. We appreciate the reviewer’s insightful suggestion to investigate whether wildtype TINF2 and the TINF2LD protein differ in activity under stress; however, this falls beyond the scope of our current study.
Comments 3: The study lacks direct evidence showing whether TIN2S retains interactions with shelterin components at a comparable level to TIN2L. Perform co-immunoprecipitation (Co-IP) and proximity ligation assays to assess TIN2S binding efficiency with TRF1, TRF2, and TPP1.
Response 3: We appreciate the reviewer’s suggestion regarding the assessment of wildtype TINF2 and TINF2LD protein binding efficiency with shelterin components. Unfortunately, due to the current unavailability of suitable antibodies for mouse shelterin proteins, conducting co-immunoprecipitation Co-IP or proximity ligation assays (PLA) is not feasible in the short term.
Comments 4: Since telomere dysfunction often manifests over time, long-term phenotypic effects should be assessed. Monitor telomere shortening and cellular senescence markers in aged TINF2S-expressing mice.
Response 4: Due to the absence of notable cellular phenotypes (i.e., regarding telomere protection and length regulation), organismal abnormalities, and lifespan changes in our genetically modified rodent models, we are currently generating aged mouse with Tinf2 LD allele and telomerase knockout background. This is a follow-up project, independent of the current study, aimed at further dissecting the roles of TINF2 variants in telomere maintenance. As these experiments are ongoing and require considerable time for completion, we apologized that we are unable to provide answers to the related questions at this time.
Comments 5: Some Western blot images (Figure 2D) appear low resolution and need better quantification (e.g., densitometry analysis).
Response 5: We have replaced the figure with a higher-resolution version and clarified the identity of protein bands (~55 kDa in Tinf2+/+ and Tinf2LD/+; ~40 kDa in Tinf2LD/+ and Tinf2LD/LD).
Comments 6: Statistical analysis should include effect size and confidence intervals, not just p-values.
Response 6: We have included effect sizes (Cohen’s d), mean differences, and 95% confidence intervals in Supplementary Table 2.
Comments 7: The manuscript states that alternative splicing is absent in Mus musculus but present in Mus spretus. Given the evolutionary importance, the authors should discuss whether forced alternative splicing of TINF2 in Mus musculus alters telomere dynamics.
Response 7: We appreciate this suggestion regarding the evolutionary significance of alternative splicing in Mus musculus versus Mus spretus. We have addressed this point in the Discussion (Line 317-318).
Comments 8: Some critical studies on TIN2 isoform function in humans (e.g., de Lange, 2018) are missing and should be cited.
Response 8: We have confirmed that all relevant TIN2 isoforms studies are cited, including 8 papers from de Lange’s lab (Reverence 8, 10, 12, 14, 15, 26, 29, 40). However, we were unable to locate a relevant study published by de Lange’s lab in 2018.

Round 2
Reviewer 2 Report
Comments and Suggestions for Authors
The authors have addressed most of the concerns raised previously. The manuscript should be acceptable for publication now.